# Endoscopic Ultrasonography Findings of Early and Suspected Early Chronic Pancreatitis

**DOI:** 10.3390/diagnostics10121018

**Published:** 2020-11-27

**Authors:** Yusuke Takasaki, Shigeto Ishii, Toshio Fujisawa, Mako Ushio, Sho Takahashi, Wataru Yamagata, Koichi Ito, Akinori Suzuki, Kazushige Ochiai, Ko Tomishima, Hiroaki Saito, Hiroyuki Isayama

**Affiliations:** Department of Gastroenterology, University of Juntendo, Tokyo 113-8421, Japan; ytakasa@juntendo.ac.jp (Y.T.); sishii@juntendo.ac.jp (S.I.); t-fujisawa@juntendo.ac.jp (T.F.); m-ushio@juntendo.ac.jp (M.U.); sho-takahashi@juntendo.ac.jp (S.T.); w.yamagata.mx@juntendo.ac.jp (W.Y.); kitoh@juntendo.ac.jp (K.I.); suzukia@juntendo.ac.jp (A.S.); k.ochiai.qd@juntendo.ac.jp (K.O.); tomishim@juntendo.ac.jp (K.T.); hiloaki@juntendo.ac.jp (H.S.)

**Keywords:** early CP, EUS, rosemont criteria

## Abstract

Chronic pancreatitis (CP) is associated with a risk of pancreatic cancer and is characterized by irreversible morphological changes, fibrosis, calcification, and exocrine and endocrine insufficiency. CP is a progressive disease with a poor prognosis and is typically diagnosed at an advanced stage. The Japan Pancreas Society proposed criteria for early CP in 2009, and their usefulness has been reported. Recently, a mechanism definition was proposed by the International Consensus Guidelines and early CP was defined as a disease state that is not based on disease duration. CP is diagnosed by computed tomography, magnetic resonance imaging, and endoscopic cholangiopancreatography, which can detect calcification and dilation of the pancreatic ducts; however, detecting early CP with these modalities is difficult because subtle changes in early CP occur before established CP or end-stage CP. Endoscopic ultrasonography (EUS) is useful in the diagnosis of early CP because it allows high-resolution, close-up observation of the pancreas. In addition to imaging findings, EUS with elastography enables measurement of the stiffness of the pancreas, an objective diagnostic measure. Understanding the EUS findings of early CP is important because a histological diagnosis is problematic, and other modalities are not capable of detecting subtle changes in early CP.

## 1. Introduction

Chronic pancreatitis (CP) causes irreversible changes such as parenchymal atrophy, fibrosis, and calcification. The complications of advanced CP include severe chronic pain and exocrine/endocrine insufficiency. Patients typically describe their pain as a dull, sharp or nagging sensation in the upper abdomen, which can radiate to the back, and often presents after, or is worsened by food intake [1,2]. Exocrine insufficiency causes symptoms such as bloating, steatorrhea and diarrhea, and requires appropriate dosage of pancreatic enzyme replacement therapy and monitoring for osteoporosis. Endocrine dysfunction often requires insulin replacement therapy and collaboration with a diabetes physician. These symptoms impair the quality of life [3]. Additionally, the incidence of pancreatic cancer is elevated in patients with CP. The standardized incidence of pancreatic cancer, adjusted for morbidity, age, and sex is 13.3–26.7 among patients with CP [4,5], and the hazard ratio for death from pancreatic cancer is 6.9 [6]. However, a definitive diagnosis of CP is typically only possible too late in the disease course to initiate treatments that might limit progression and/or minimize complications. To avoid advanced-stage complications and improve clinical outcomes, early diagnosis and treatment are essential before CP becomes established and irreversible [7]. Diagnosis of early CP is clinically challenging due to the lack of sensitive and specific methods or a gold standard technique. CP is diagnosed by the presence of stones in the pancreatic ducts or calcifications throughout the pancreas by computed tomography (CT), X-ray, or abdominal ultrasonography (AUS). It can also be diagnosed by irregular dilatation of the main pancreatic duct and irregular dilatation of pancreatic duct branches with variable intensity and a scattered distribution via endoscopic cholangiopancreatography (ERCP) and magnetic resonance cholangiopancreatography (MRCP). The Cambridge classification [8] incorporated CT, AUS and ERCP features to classify and grade disease severity. These diagnostic modalities accurately identify patients without pancreatic pathologic conditions and those with severe CP. However, the Cambridge classification provided poor diagnostic accuracy in evaluation of patients with equivocal or early CP. Such definitive image findings are absent in early CP, which cannot be diagnosed by those modalities. Endoscopic ultrasonography (EUS) enables detailed examination and scoring of the pancreatic parenchyma and duct because of its high resolution, close-up observation of the pancreas [9,10]. Therefore, EUS may be able to identify subtle changes in patients indicating early CP [11]. Severity, as observed using EUS, is classified based on the number of positive findings, and advanced CP with calcification can be classified as mild depending on the number of findings. This could result in a discrepancy between clinical progression and the EUS severity classification [12]. Therefore, EUS-based diagnostic criteria for CP were determined by expert consensus and based on the available evidence. The Rosemont criteria (RC) were published in 2009 [13] (Table 1).

The RC categorizes EUS findings as major A, major B, and minor, such that diagnosis is not only based on the number of positive findings. The major RC are hyperechoic foci with shadowing, main pancreatic duct (MPD) calculi, and lobularity with honeycombing. The minor criteria comprise cysts, strands, non-shadowing hyperechoic foci, lobularity without honeycombing, an MPD dilated > 3.5 mm, irregular MPD contour, side branches dilated > 1 mm, and a hyperechoic MPD margin. According to the RC, each case is classified as consistent with CP, suggestive of CP, indeterminate for CP, or normal. The RC do not encompass early CP, but EUS findings—as described in the RC—have been used to investigate early CP. The ability of EUS to diagnose early CP has been evaluated [14,15,16,17,18,19]. However, accurate diagnosis of early CP is not possible using imaging criteria alone. EUS also enables indirect assessment of pancreatic exocrine insufficiency [20] and is useful for indicating treatment for both CP and early CP.

## 2. Materials and Methods

### 2.1. EUS Findings of Early CP

According to EUS, normal pancreatic parenchyma has similar or slightly higher echogenicity than the liver and exhibits a fine reticulate pattern [21]. Although few papers have defined imaging findings for early CP, the Japan Pancreas Society (JPS) proposed imaging findings for early CP based on RC. The EUS findings of early CP are of the minor and major B categories of the RC. However, hyperechoic foci with shadowing, MPD dilation, and irregular MPD contour are excluded because these findings are seen in established or end-stage CP. The histopathological changes reflected by the EUS findings of early CP are unclear due to the paucity of surgical studies and the scarcity of EUS-guided fine-needle aspiration specimens, which hampers evaluation. Histologic correlations among the EUS findings for RC were obtained for cyst, hyperechoic ductal margins, and dilated side branches. A cyst is considered to represent a pseudocyst or retention cyst, hyperechoic ductal margin to indicate fibrosis of the pancreatic duct, and dilated side branches to represent side-branch ectasia. Albshir et al. [22] reported that the number of preoperative EUS findings was significantly correlated with the number of fibrosis findings in surgical specimens. Varadarajulu et al. [23] performed a prospective study of the correlations between EUS criteria as they pertain to non-calcific CP and the histology results of surgical specimens. The preoperative EUS findings were significantly associated with those of resected pancreatic tissue. These reports suggest that EUS findings can be associated with fibrosis of the pancreatic parenchyma.

1.Lobularity (Figure 1a)

Lobularity is defined endosonographically as a reticulated area surrounded by ≥5 mm structures with rims that are hyperechoic relative to the echogenicity in the central areas. At least three lobules in the body or tail are necessary for the feature to be considered present. If at least three of the lobules are contiguous, the feature is termed “lobularity with honeycomb” according to the RC, a major B criterion. If lobules are non-contiguous, the feature is termed “lobularity without honeycomb”, a minor RC. Lobularity with honeycomb is considered to indicate stronger CP changes than lobularity without honeycomb, but the 2019 JPS criteria make no such distinction. Lobularity should be assessed only within the pancreatic body and tail because this finding is relatively common in the pancreatic head, especially the ventral pancreatic area.

2.Hyperechoic foci without shadowing (Figure 1b)

Hyperechoic foci with post-acoustic shadowing indicate intraparenchymal calcification and are indicative of a major A on the RC. This finding is considered as established CP. Hyperechoic foci without shadowing are defined as echogenic structures ≥3 mm in length and width with no shadowing. This finding is classified as minor according to the RC. At least three of these structures are needed for the feature to be considered abnormal. Hyperechoic foci without shadowing are indicative of focal fibrosis.

3.Strands (Figure 1c)

Strands are defined as hyperechoic lines ≥3 mm in length in at least two directions with respect to the imaged plane. At least three of these structures are needed for the feature to be considered abnormal. This finding is indicative of bridging fibrosis.

4.Hyperechoic MPD margin (Figure 1d)

A hyperechoic MPD margin is defined as an echogenic, distinct structure accounting for > 50% of the entire MPD in the pancreatic body and tail. Evaluation is subjective because it is difficult to assess the pancreatic ducts along the long axis of linear-type EUS. This finding is indicative of a thickened pancreatic duct wall [12].

5.Dilated side branches (Figure 1e)

Dilated side branches are defined as three or more tubular echoic structures each ≥1 mm in width and budding from the MPD. This finding is thought to be caused by narrowing of the branches due to micro-fibrosis.

It is difficult to determine whether the above findings are indicative of early CP or changes due to aging. A study on the association between EUS findings of CP and age reported a trend in which abnormal EUS findings increase with age in patients over 60 years of age [24]. Whether these findings are indicative of early CP or aging changes should be determined clinically.

### 2.2. Definition and Assessment of Early CP

The term early stage CP was first used by Ammann et al. in 1996 [25]. They argued that the definition of alcoholic early CP should be independent of histological findings such as those from long-term follow-up that would eventually reveal the typical clinical features of CP [25]. Before using early stage CP, the majority of classification systems and consensus guidelines applied the term probable CP for cases with a high likelihood of CP but that did not meet the imaging criteria for definite CP. The American Pancreatic Association (APA) published practice guidelines for CP in 2014 [26]. These guidelines suggested classification of CP based on imaging or histological findings, but early CP could not be defined due to insufficient evidence. The United European Gastroenterology evidence-based guidelines for the diagnosis and therapy of CP were published in 2017 [27]. These guidelines suggested classification of CP based on imaging or histological findings, but early CP could not be defined due to insufficient evidence. The United European Gastroenterology evidence-based guidelines for the diagnosis and therapy of CP were published in 2017 [20]. In 2016, The International Consensus Guidelines on CP (formulated by the International Association of Pancreatology, APA, JPS, and European Pancreatic Club) were proposed [28]. Those consensus guidelines were based on the mechanistic definitions “CP is a pathologic fibro-inflammatory syndrome of the pancreas in individuals with genetic, environmental and/or other risk factors who develop persistent pathologic responses to parenchymal injury or stress” and “common features of established and advanced CP included pancreatic atrophy, fibrosis, pain syndromes, duct distortion and strictures, calcifications, pancreatic exocrine dysfunction, pancreatic endocrine dysfunction and dysplasia [28]. The mechanistic definitions offer a potential solution to the problem of defining and diagnosing early CP and provide a framework for diagnostic criteria by considering risk factors, biomarkers of inflammation, pain, and functional status within the clinical context. The consensus guidelines adopt the progressive model of CP consisting of five disease stages: at risk, acute pancreatitis (AP)—recurrent AP, early CP, established CP, and end-stage CP. A recent international consensus on early CP failed to agree on diagnostic criteria [29]. However, a consensus was achieved on defining “early” as a disease state that is not based on disease duration, because early CP cannot be diagnosed by imaging modalities alone. Although no international consensus was reached, the concept of early CP is very important. Because an understanding of CP based on the mechanism definition is a fundamental concept and an enhanced understanding of CP imaging findings, particularly those of EUS, is required. There is a need to decide on a definition of early CP that allows us to reach an international consensus.

### 2.3. Japanese Diagnostic Criteria for Early CP

In 2009, the first diagnostic criteria for early CP were proposed by the JPS [7]. Early CP was defined as the absence of CP findings and the presence of at least two of the four clinical and imaging findings of early CP upon EUS or ERCP. Based on the RC [13], the EUS findings of early CP are as follows (five parenchymal and two ductal features): (1) lobularity with honeycombing, (2) lobularity without honeycombing, (3) hyperechoic foci without shadowing, (4) stranding, (5) cysts, (6) dilated side branches, and (7) hyperechoic MPD. More than two of the seven features are sufficient for the diagnosis of early CP. Among 83 patients diagnosed with early CP using the JPS criteria and who completed a 2-year follow-up, four (4.8%) progressed to defined CP. All four patients were male, had a history of drinking alcohol and smoking, and continued to drink alcohol. By contrast, 31 patients (37.3%) were downgraded [30]. Additionally, Hashimoto et al. [31] reported that epigastric pain syndrome and pancreatic enzyme abnormalities were associated with early CP in JPS criteria. Yamawaki et al. [32] reported that Camostat Mesilate, Pancrelipase, and Rabeprazole combination therapy improved the epigastric pain associated with early CP. Thus, the JPS criteria can be used to identify early CP. However, in some patients, the EUS findings improved without treatment. Sheel et al. [33] followed up 66 patients previously diagnosed with CP by EUS. Over a median of 3 years of follow-up, one-third of the patients with EUS features of early CP progressed to definite CP. This suggests that the EUS imaging criteria have low specificity. Therefore, the JPS CP criteria were revised in 2019 to increase their diagnostic specificity [34]. The 2019 CP criteria included two additional factors—genetic mutations and history of acute pancreatitis—and a revised definition of continuous heavy alcohol drinking as ≥60 g/day of pure ethanol to be consistent with conditions of alcoholic liver injury. New diagnostic criteria for pancreatitis-related genetic abnormalities and history of acute pancreatitis have been added to the diagnostic criteria, with emphasis on assessment of risk factors based on the mechanism definition. Recently, SPINK1 and PRSS gene abnormalities have been reported to be associated with hereditary pancreatitis or idiopathic CP [35,36,37,38,39,40], and these two gene abnormalities have been added to the list of items in this revision. The JPS 2019 CP criteria comprise seven items: (1) characteristic imaging findings, (2) characteristic histological findings, (3) repeated upper abdominal pain, (4) abnormal pancreatic enzyme levels in the serum or urine, (5) abnormal pancreatic exocrine function, and (6) continuous heavy drinking of alcohol equivalent to or more than 60 g/day of pure ethanol or a genetic mutation (PRSS1 or SPINK1), and (7) history of acute pancreatitis (Table 2).

The characteristic imaging findings have also been revised. MRCP can be used for initial screening because of recent improvements in its resolution. Ito et al. [41] reported that irregular dilatation of three branches on MRCP was more often observed in early CP, defined by JPS criteria 2009, compared to non-early CP. The EUS findings were revised to be simpler because of problems with inter-observer agreement rates [42,43,44]. The EUS findings are as follows: (1) hyperechoic foci (non-shadowing) or strands, (2) lobularity, (3) hyperechoic MPD margin, and (4) dilated side branches. At least two of the four EUS findings, including (1) or (2), are required for the diagnosis of early CP. The presence of cysts was excluded because it can be difficult to distinguish pancreatic cysts such as intraductal papillary mucinous neoplasm. A diagnostic flowchart for CP created by the author based on the 2019 JPS CP criteria is shown in Figure 2.

### 2.4. Limitations of the Diagnostic Criteria for Early CP

The JPS criteria were the first to refer to the diagnosis of early CP, and it is expected that revision will further increase their accuracy. However, an international consensus definition of early CP has yet to be reached due to several limitations. Some patients diagnosed with early CP progress to CP and others do not. This may be due to the low accuracy of the diagnostic criteria for early CP. Another problem is the low inter-observer agreement rate of EUS findings for CP. Koh et al. [44] reported a multicenter validation study in Asia. In this study, the inter-observer agreement rates were determined for “Hyperechoic foci with shadowing”, “Non-shadowing hyperechoic foci”, “Strands”, “Lobularity with non-contiguous lobules”, “Lobularity with honeycombing”, “Cysts”, “Dilated main pancreatic duct”, “Irregular pancreatic duct”, “Hyperechoic duct wall”, “Dilated side branches”, “Calculi in the main pancreatic duct”. Cysts and the calculi in the main pancreatic duct had a kappa > 0.8, five factors (hyperechoic foci with shadowing, lobularity with honeycombing, dilated main pancreatic duct, irregular pancreatic duct, dilated side branches) had a kappa of 0.6–0.8, and four factors (non-shadowing hyperechoic foci, strands, lobularity with non-contiguous lobules, hyperechoic duct wall) had a kappa < 0.6. Diagnostic criteria consist of clinical, hematological, and imaging findings. Some patients may meet some, but not all, of the diagnostic criteria, and those are classified as suspected early CP. The concept of early CP or suspected early CP is shown in Figure 3.

The natural history of early CP and whether it requires for treatment are unclear, and so further research is needed. Genetic factors are also important in patients with CP. However, evidence for a genetic factor in early CP is lacking. In addition, genetic and complicated pancreatic exocrine function tests may not be performed in clinical practice. Additionally, patients with idiopathic CP may have not been adequately evaluated because of the disproportionately large weight assigned to continuous heavy alcohol drinking as a clinical/functional feature in the diagnostic criteria.

### 2.5. EUS-Elastography

EUS is the most sensitive imaging technique for the diagnosis of CP, mainly during the early stages. However, objective measures are required because of the potential for low inter-observer agreement. Elastography evaluates tissue hardness and is based on the principle that softer parts of tissues more readily deform under compression than harder parts. EUS-elastography enables the visualization of tissue stiffness, including that of the pancreatic parenchyma. There are two types of elastography: qualitative and semi-quantitative/quantitative [45]. Strain elastography is a qualitative method. This technique measures strain, which is negatively correlated with tissue hardness and allows the direct visualization of information reflecting strain as a strain distribution map (elastogram), which, for visualization purposes, is color-coded and displayed next to the conventional B-mode image on the screen. To improve the accuracy and reproducibility of the elastography and to reduce the human bias, semiquantitative/quantitative analysis of tissue stiffness has been used. Semi-quantitative elastography analysis include strain histogram (SH) and strain ratio (SR), and quantitative elastography analysis includes shear-wave elastography. EUS-strain elastography includes methods based on region of interest (ROI) coloration and image heterogeneity and measurement of the SR for image quantification. The value of EUS-strain elastography for the diagnosis of CP based on parenchymal stiffness [46] and pancreatic exocrine insufficiency [47,48] has been discussed. Kim et al. reported a mean SR of 3.78 ± 1.35 for the normal pancreas and 8.21 ± 5.16 for CP; the sensitivity, specificity, and accuracy of the SR for detecting CP using a cut-off value of 5.62 were 71.6%, 75.2%, and 74.8%, respectively [49]. The results of EUS-strain elastography are correlated with CP stage according to the RC [50]. EUS-strain elastography is expected to improve the accuracy of CP diagnosis but has several limitations. First, it is an operator-dependent procedure that does not measure an absolute hardness value. Second, EUS-strain elastography is affected by the size and/or position of the ROI. EUS-shear-wave elastography exhibits greater precision for CP cases because it measures the absolute hardness value of the pancreatic parenchyma [51,52]. An acoustic radiation force (push pulse) is sent to the focal point of the ROI, at the edge of which a shear wave is generated. The velocity of the shear wave between two search points is calculated with a tracking pulse. Yamashita et al. [53] reported that the diagnostic accuracy of EUS-shear-wave elastography for CP was 0.97, and based on a cut off value of 2.19, the modality exhibited 100% sensitivity and 94% specificity for the diagnosis of CP. However, there was no difference between patients with indeterminate CP according to the RC (which is considered equivalent to early CP) and non-CP patients. The limitations of EUS-shear-wave elastography include the difficulty in evaluating a single location in the pancreas, its usage difficulty in obese patients, and instability caused by breathing. The usefulness of EUS-elastography for the diagnosis of early CP is unknown; therefore, further studies are needed. EUS-elastography, which is minimally invasive and enables quantitative evaluation, will facilitate the diagnosis of early CP.

### 2.6. The Future of Diagnosis in Early CP

Imaging diagnosis of early CP requires sensitive diagnostic modalities to capture subtle morphological changes such as fibrosis of the pancreatic parenchyma and dilation of the branch, therefore, EUS is considered to be the most suitable modality. Secretin-enhanced MRCP, contrast-enhanced AUS, and non-contrast spin-label MRI have been investigated as methods to evaluate the early CP. A recent study, which specified Magnetic resonance Imaging as a Non-Invasive Method for the Assessment of Pancreatic fibrosis (MINIMAP), was designed to incorporate both the parenchymal and ductal features of chronic pancreatitis [54]. These methods have the potential to be useful in the diagnosis of early CP. The modalities for imaging of early CP required are minimally invasive, simple and sensitive. Furthermore, it is important to be able to diagnose early CP in a combination of diagnostic imaging and the development of laboratory tests, which can measure the decline of early endocrine and exocrine functions. Artificial intelligence (AI) is also useful for the diagnosis of early CP. In EUS, AI is used for detecting anatomical features, differential pancreatic tumors and cysts [55,56]. AI systems and deep learning based on neural networks, which is a machine learning technique that is widely used in the medical field, involves three phases. First, data are collected and annotated, second, the deep learning architecture is built and third, training and ability validation takes place. There is no consensus on the definition of early chronic pancreatitis, so AI is not yet needed, but it may solve the problem of interobserver agreement in the future.

## 3. Conclusions

CP is not curable, and most patients will experience longstanding symptoms, such as bloating, steatorrhea and diarrhea, which detract from their quality of life. Patients with established CP or end-stage CP will likely eventually require treatment for abdominal pain, exocrine insufficiency with maldigestion, and endocrine insufficiency with diabetes. Treatment and lifestyle changes should be recommended to patients with early CP before their disease progresses. EUS is important for the diagnosis of early CP. Some findings can be detected only by EUS, and its combination with, for instance, elastography and contrast studies will further improve the accuracy of diagnosis of early CP. The mechanistic definition proposed a concept of early CP, which can exhibit the reversible structural fibrotic change of end stage CP. However, no international consensus has been reached on its definition. Further advances in imaging devices may reveal the pathogenesis of early CP, just as EUS has gone from mechanical to electronic with increased resolution to provide more detailed information. Future technologies such as AI may also improve the accuracy of diagnosis of early CP. Such advances may lead to an international consensus, although the diagnosis of early CP is currently controversial.

## Figures and Tables

**Figure 1 diagnostics-10-01018-f001:**
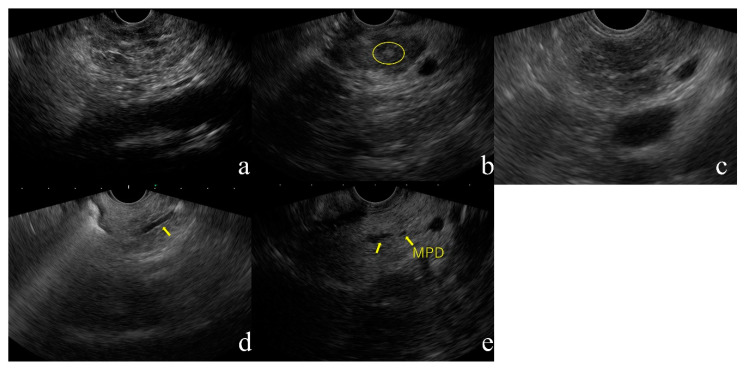
Diagnostic flowchart of CP in JPS guideline 2019. Abbreviations, CP, chronic pancreatitis; JPS, Japan Pancreas Society. (**a**) Lobularity, (**b**) Hyperechoic foci without shadowing, (**c**) Strands, (**d**) Hyperechoic MPD margin, (**e**) Dilated side branches

**Figure 2 diagnostics-10-01018-f002:**
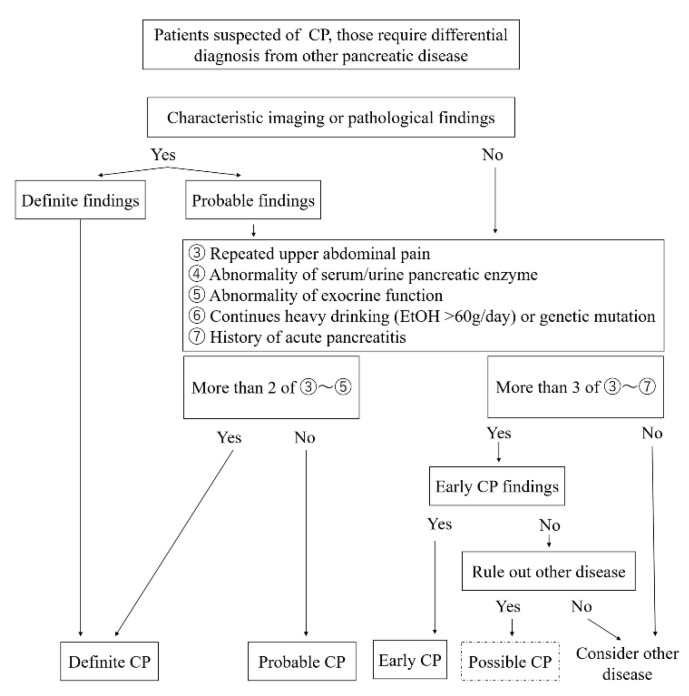
Endoscopic ultrasonography (EUS) images of the CP based on JPS guideline.

**Figure 3 diagnostics-10-01018-f003:**
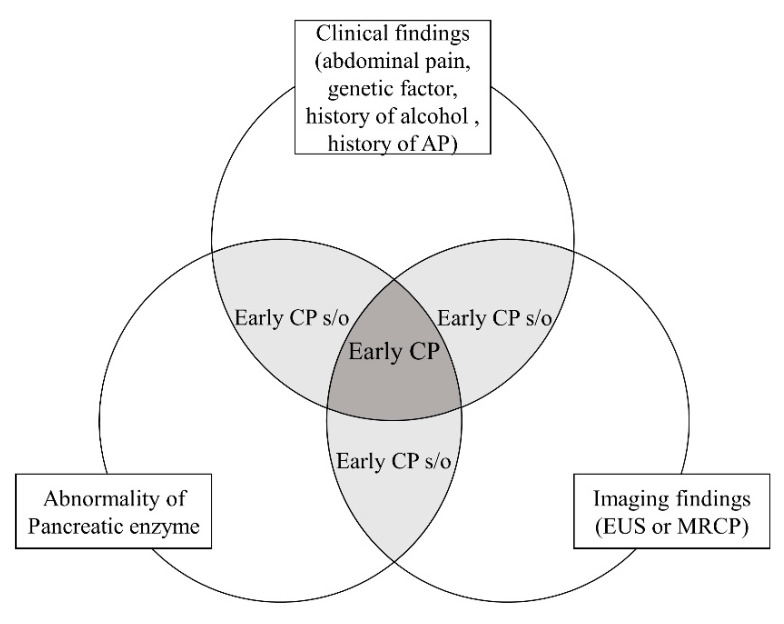
The concept of early CP.

**Table 1 diagnostics-10-01018-t001:** The Rosemont classification.

Parenchymal Criteria
Major A
Hyperechoic foci (>2 mm in length/width with shadowing)
Major B
Lobularity (≥3 contiguous lobules = “with honeycombing”)
Minor
Cyst (anechoic, round/elliptical with or without septations)
Hyperechoic strands (≥3 mm in at least 2 different directions with respect to the imaged plane)
Hyperechoic foci (>2 mm in length/width with no shadowing) Lobularity (noncontiguous lobules = “without honeycombing”)
**Duct Criteria**
Major A
MPD calculi (echogenic structure(s) within the MPD with acoustic shadowing)
Minor
MPD dilation (≥3.5 mm in body or >1.5 mm in tail)
Irregular MPD contour (uneven or irregular outline and ectatic course)
Dilated side branches (>3 tubular anechoic structures each measuring ≥1 mm in width, budding from the MPD)
Hyperechoic MPD margin (echogenic, distinct structure >50% of entire MPD in the body and tail)

EUS, endoscopic ultrasonography; MPD, main pancreatic duct.

**Table 2 diagnostics-10-01018-t002:** Clinical diagnostic criteria for CP in 2019 by JPS.

Diagnostic Items for CP
(1) Characteristic imaging findings
(2) Characteristic histological findings
(3) Repeated upper abdominal pain
(4) Abnormal pancreatic enzyme levels in the serum or urine
(5) Abnormal pancreatic exocrine function
(6) Continuous heavy drinking of alcohol equivalent to 60 g/day of pure ethanol or genetic mutation (PRSS1 or SPINK1)
(7) History of acute pancreatitis
Definite CP: either a or b
a. Definite findings of (1) or (2)
b. Probable findings of (1) or (2), plus more than two items among (3), (4), and (5)
Probable CP
Probable findings of (1) or (2)
Early CP
More than three items among (3)–(7) plus image findings of early CP

JPS, the Japan Pancreas Society; CP, chronic pancreatitis.

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
