# Peer review of "Endoscopic Ultrasonography Findings of Early and Suspected Early Chronic Pancreatitis"

_diagnostics, 2020, doi:10.3390/diagnostics10121018_

Round 1

Reviewer 1 Report

This paper is very well written. I think it provides useful information to the reader. There are no corrections or comments.

Author Response

We wish to thank the reviewer for comment. We made a minor revision in accordance with the another reviewer comments.

Reviewer 2 Report

Authors described in a review the role of EUS to detect early and suspected early pancreatitis and its importance for an early diagnosis to prevent damages on parenchyma and clinical status of the patient.

However some revisions are needed, in particular:

  • Figure 1 needs revisions, especially in order to provide a clear graphic explanation about  hyperechoic foci without shadowing (fig. 1b) and strands (fig. 1c). Please, add arrows to signal in fig.d and fig.e the hyperechoic walls of main pancreatic duct anche parechymal cysts in early chronic pancretitis.
  • Sentence in lines 166-170 is quite convoluted and difficult to be understood. Please revised it in order to offer a better explanation.
  • Elastography session needs a revision, particularly in the initial part. Please, added specific references and correct the sentence in lines 248-249 about elastography types. There are two types of elastography: qualitative (and sub-types) and semi-quantitative/quantitative (sub-types). Please, correct it.

Author Response

Response: We wish to express our appreciation to the reviewers for their insightful comments on our paper. The comments have helped us significantly improve the paper.

Point 1: In accordance with the reviewer's comment, we have revised Figure1. First, we have added the circle to the hyperechoic foci in Fig1-b for clarity, and second, exchanged the image to clearer image in Fig1-c. Third, we have added the arrow in Fig1-d and Fig1-e.

Point2: We have revised the sentence in lines 166-170 as follow.

“Although no international consensus was reached, the concept of early CP is very important. because an understanding of CP based on the mechanism definition is a fundamental concept and an enhanced understanding of CP imaging findings, particularly those of EUS, is required. There is a need to make a definition of early CP that allows us to reach an international consensus.”

“The definition of early CP was not reached to international consensus, because Morphology based diagnosis of Early CP is not possible without additional information. EUS is important but not sufficient for early CP diagnosis. In the statement, early CP cannot be diagnosed based on currently available imaging techniques alone. Based on the mechanism definition, new diagnostic criteria will be needed that combines factors such as environmental and genetic factors with EUS findings.”

Point3: We have revised the Elastography section and add the reference as follow.

“There are two types of EUS-elastography: strain elastography and shear-wave elastography.”

“There are two types of elastography: qualitative and semi-quantitative/quantitative (45). Strain elastography is a qualitative method. This technique measures strain, which is negatively correlated with tissue hardness and allows the direct visualization of information reflecting strain as a strain distribution map (elastogram), which, for visualization purposes, is color‑coded and displayed next to the conventional B‑mode image on the screen. To improve the accuracy and reproducibility of the elastography and to reduce the human bias, semiquantitative/quantitative analysis of tissue stiffness has been used. Semi‑quantitative elastography analysis include strain histogram (SH) and strain ratio (SR), and quantitative elastography analysis include shear-wave elastography.”
